# Allosteric control of olefin isomerization kinetics via remote metal binding and its mechanochemical analysis

Yichen Yu[1,3], Robert T. O'Neill [2,3], Roman Boulatov [2] ✉,
Ross A. Widenhoefer[1] ✉ & Stephen L. Craig [1] ✉

Allosteric control of reaction thermodynamics is well understood, but the mechanisms by which changes in local geometries of receptor sites lower activation reaction barriers in electronically uncoupled, remote reaction moieties remain relatively unexplored. Here we report a molecular scaffold in which the rate of thermal E-to-Z isomerization of an alkene increases by a factor of as much as $10^4$ in response to fast binding of a metal ion to a remote receptor site. A mechanochemical model of the olefin coupled to a compressive harmonic spring reproduces the observed acceleration quantitatively, adding the studied isomerization to the very few reactions demonstrated to be sensitive to extrinsic compressive force. The work validates experimentally the generalization of mechanochemical kinetics to compressive loads and demonstrates that the formalism of force-coupled reactivity offers a productive framework for the quantitative analysis of the molecular basis of allosteric control of reaction kinetics. Important differences in the effects of compressive vs. tensile force on the kinetic stabilities of molecules are discussed.

The central role of allosteric regulation in enabling life as we know it[1] and the fundamental questions in information transfer[2], conformational dynamics[3,4] and emergent properties[5,6] that allostery presents underlie continued effort to design synthetic molecules with allosteric behavior[7]. Allosteric control of reaction thermodynamics, particularly binding affinities of small ligands, has been realized in thousands of synthetic molecules, and the design principles of such receptors are well understood[8–12].

In comparison, abiological examples of allosteric control of reaction kinetics are few[8,12]. The most productive approach to date has been to combine a known catalyst with a regulating moiety for on/off control, with the catalyst in the on state retaining largely unchanged the properties of the non-allosteric equivalent. Despite clever applications of such catalysts[13,14], they are thought to offer only limited insights into how electronically uncoupled changes in local geometries of remote receptor sites lower activation reaction barriers below those of non-allosteric equivalents[8].

Here we report a molecular scaffold which allows the rate of thermal E→Z isomerization of an alkene (stiff stilbene[15], Fig. 1) to vary systematically by a factor of up to $10^4$ in response to subtle structural perturbations triggered by fast binding of a metal ion to a remote receptor site. Representing the latter as a harmonic compressive potential constraining E stiff stilbene, which metal binding increases, reproduced measured kinetics accurately. Our work provides an experimentally validated generalization of mechanochemical kinetics to compressive loads[16], and it demonstrates that the formalism of force-coupled reactivity constitutes a productive and quantitative approach to the analysis of allosteric control of reaction kinetics. We are unaware of previous use of a mechanochemical formalism to support quantitative analysis of allosteric accelerations, which here

[1]Department of Chemistry, Duke University, Durham, NC 27708, USA. [2]Department of Chemistry, University of Liverpool, Crown Street, Liverpool L69 7ZD, UK. [3]These authors contributed equally: Yichen Yu, Robert T. O'Neill. ✉e-mail: boulatov@liverpool.ac.uk; ross.widenhoefer@duke.edu; stephen.craig@duke.edu

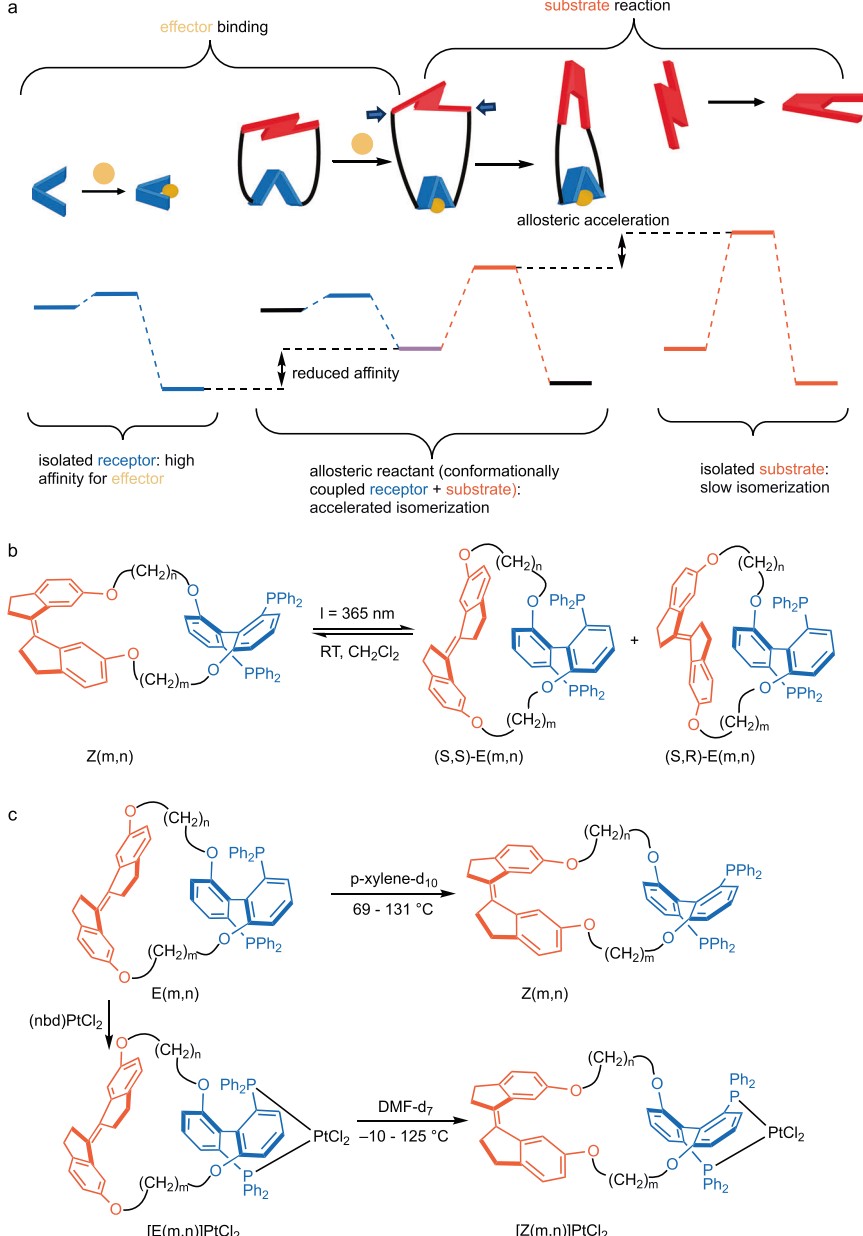

**Fig. 1 | The concept of allosteric reaction control and its implementation. a** A substrate reaction is accelerated allosterically if it is coupled to a remote receptor site such that binding of another molecule (effector) to this site suitably strains the substrate (in this example by imposing a compressive load, represented by a pair of square arrows, on the substrate that contracts during the reaction). This strain comes at the expense of reduced affinity of the effector for the receptor. **b, c** Here, allosteric acceleration of E-stiff stilbene isomerization (red) by binding of PtCl₂ moiety to a bidentate phosphine receptor (blue) is demonstrated using a series of strained macrocycles, E(m,n). These are synthesized by photoisomerization of

strain-free Z(m,n) analogs (**b**); the two diastereomers of Z macrocycles are in rapid equilibrium and only the S configuration of the stiff stilbene is shown; conversely, epimerization of *E*-stiff stilbene in macrocycles is negligibly slow. Photo-isomerization of Z(3,3) yields a separable mixture of (*S,S*)-E(3,3) and (*S,R*)-E(3,3) diastereomers which differ in the configuration of *E*-stiff stilbene; (*S,R*)-E(2,3) and (*S,R*)-E(2,2) are formed as single diastereomers. In the presence of a source of the effector (nbd = norbornadiene), rapid coordination of PtCl₂ to the phosphine on the opposite side of *E*-stiff stilbene accelerate its thermal isomerization of the Z isomer (**c**), as illustrated on the example of the *S,S*-diastereomer.

provides insight into how molecular structure enables allostery. The work also expands the very limited range of reactions demonstrated to be sensitive to externally imposed compressive force[17], and highlights important differences in the effects of compressive vs. tensile force on the kinetic stabilities of molecules.

## Results and discussion

We studied strained macrocycles comprising an (*S*)-BIPHEP moiety as the effector binding site (blue, Fig. 1) that bridges and constrains the separation of the C6,C6′ atoms of *E*-stiff stilbene (the substrate, red) on

the opposite side of the macrocycle. Strain within the macrocycle is modulated by the length of the alkyl chains that tether the BIPHEP and *E*-stiff stilbene. Following the previously reported protocol[17,18], we generated the strained E(m,n) macrocycles by photoisomerization of the relaxed Z(m,n) macrocycles (Fig. 1b): the axial chirality of both the stiff-stilbene and BIPHEP moieties makes all macrocycles diastereomeric. Z(3,3) yielded a separable mixture of (*S,S*)-E(3,3) and (*S,R*)-E(3,3) diastereomers, whose diselenide derivatives were previously characterized by X-ray crystallography[19], whereas smaller macrocycles, E(2,2) and E(2,3) were generated as single diastereomers as evidenced

**Table 1 | Summary of isomerization kinetics**

| macrocycle | T (°C) | $\Delta G^{\ddagger}_{expt}$[a,b] | $\Delta G^{\ddagger}_{calc}$[c] |
|---|---|---|---|
| (S,R)-E(2,2) | 69–84 | 27.0 ± 2.0 | 26.1*, 23.7 |
| [(S,R)-E(2,2)]PtCl₂ | −10–6 | 20.3 ± 2.0 | 18.7*, 18.0 |
| (S,R)-E(2,3) | 120–135 | 31.0 ± 2.0 | 29.6*, 30.4 |
| [(S,R)-E(2,3)]PtCl₂ | 64–80 | 26.7 ± 1.9 | 26.4*, 24.7 |
| (S,S)-E(3,3) | 131 | 31.4 ± 0.1[d] | 33.8[d] |
| [(S,S)-E(3,3)]PtCl₂ | 125 | 30.0 ± 0.1[e] | 29.7[e] |
| (S,R)-E(3,3) | 128 | 31.0 ± 0.1[f] | 32.7[f] |
| [(S,R)-E(3,3)]PtCl₂ | 121 | 29.6 ± 0.1[g] | 32.1[g] |

The measured ($\Delta G^{\ddagger}_{expt}$) and calculated ($\Delta G^{\ddagger}_{calc}$) activation free energies of stiff stilbene E → Z isomerization in E(m,n) macrocycles and their platinum complexes [E(m,n)]PtCl₂ in kcal/mol.
[a]at 298 K unless noted otherwise.
[b]free ligand in p-xylene-$d_{10}$; metal complex in DMF-$d_7$.
[c]at 298 K unless specified otherwise, in vacuum at 1 atm; the two values for E(2,n), n = 2 or 3 correspond to the (S,R) and (S,S) diastereomers where * indicates the more stable diastereomer.
[d]T = 131 °C.
[e]T = 125 °C.
[f]T = 128 °C.
[g]T = 125 °C.

by ¹H NMR spectroscopy[19]. We assigned these diastereomers as (S,R), which we calculated (see below) to be less strained than the (S,S) analogs, based on the previously reported predominant generation of the least strained E diastereomers by photoisomerizations of Z-stiff stilbene macrocycles[20,21]. This preference was rationalized by a combination of rapid epimerization of Z-stiff stilbene[22] and the strong dependence of the quantum yields of Z→E photoisomerization on the strain energy of the product[23–25]. Macrocycles E(m,n) react rapidly and quantitatively with (NBD)PtCl₂ (nbd = norbornadiene) at or above −30 °C to form the ligated macrocycles [E(m,n)]PtCl₂ (Fig. 1c). The PtCl₂ fragment was chosen as effector for the high kinetic and thermodynamic stability of the Pt−P bond[26].

We measured the activation barriers of E→Z isomerization of stiff stilbene in the absence of metal ions and upon coordination to PtCl₂ (Supplementary Figs 1–24). In p-xylene-$d_{10}$ solutions at ≥69 °C the Pt-free macrocycles E(m,n) isomerized exclusively to their Z(m,n) isomers with the first-order kinetics over three half-lives as determined by ³¹P NMR spectroscopy (Table 1). The isomerization rate increases as the size of the macrocycle decreases, e.g., E(2,2) > E(2,3) > E(3,3), which follows the trend in the relative strain energy of these macrocycles, e.g., (S,R)-E(2,2), (S,R)-E(2,3) and (S,R)-E(3,3) are 19.9, 10.7 and 9.9 kcal/mol less stable than the corresponding Z isomers. However, as demonstrated previously restoring forces of carefully chosen internal molecular coordinates, rather than relative energies, are often a better quantitative correlant of reactivity[27–29]. As discussed below, the mechanochemical formalism captures quantitatively the variation in isomerization kinetics across the whole range of the macrocycles, whether metalated or metal-free. Similarly, warming solutions of platinum-ligated macrocycles [E(m,n)]PtCl₂ in DMF-$d_7$ at or above −10 °C led to first-order decay over three half-lives to form exclusively the corresponding [Z(m,n)]PtCl₂ macrocycles. We previously demonstrated[20] that the isomerization kinetics of stiff stilbene varies little with reaction solvent, a conclusion supported by our own control experiments on (S,R)-E(2,3) (Supplementary Fig. 25). Consequently, we chose the solvents for kinetic measurements that maximize the solubility of the components, e.g., xylene for Pt-free macrocycles and DMF for Pt analogs. Given the range of activation energies across the full series of macrocycles, temperatures were chosen on a case-by-case basis to achieve rates that were practical for NMR studies (Table 1). The low activation free energies of isomerization of [(S,R)-E(2,3)]PtCl₂ and [(S,R)-E(2,2)]PtCl₂ macrocycles allowed us to measure the E→Z isomerization kinetics across a range of temperatures and extrapolated those results to the standard temperature of 298 K.

In each studied macrocycle, platinum coordination lowered the activation free energies of isomerization, by between 6.7 ± 2.0 kcal/mol in the smallest (S,R)-E(2,2) macrocycle and 1.4 ± 0.1 kcal/mol in the largest E(3,3) macrocycles. These decreases in activation energies are dominated by reduced activation enthalpies [e.g., 6.8 ± 2.0 kcal/mol in (S,R)-E(2,2) and 6.0 ± 2.0 kcal/mol in (S,R)-E(2,3)], as expected for an elementary reaction with a non-polar transition state.

To understand the structural and energetic origin of the accelerated isomerization of stiff stilbene by metal coordination and to test the utility of a force-based approach to broader analysis of allosterically controlled reaction kinetics we optimized conformational ensembles of Z and E isomers of both metal-free and Pt-coordinated macrocycles at (u)BMK/def2SVP//(u)B3LYP/def2SVP level of DFT in vacuum. The calculations reproduced measured $\Delta G^{\ddagger}$ to within 1–2.5 kcal/mol (Table 1).

The optimized geometries reveal compressively strained E-stiff stilbene (SS), as evidenced by the contraction of its $_{Ar}$C-C = C-C$_{Ar}$ torsion, θ, relative to that of free E-SS (θ ≈ 180°), in all macrocycles (Fig. 2a). In all Pt complexes this torsion was reduced further by 4–8° compared to the metal-free congeners, which can be attributed to the contraction of the BiPHEP-containing strap needed to accommodate the preferred Pt-P bond distances and P-Pt-P bond angle. Conversely, the corresponding torsion angles of the conformers comprising the transition states of isomerization deviate from the strain-free value by between −6° and 2°, depending on the conformer (Supplementary Fig. 26), suggesting low strain of either tension or compression, regardless of the presence of Pt.

Across all conformers, geometric parameters of the SS(OCH₂)₂ moieties (indicated in blue, red and green in Fig. 2a) closely resemble those of bis-(6,6'-dimethoxy)stiff stilbene, SS(OMe)₂, with a compressive force of varying magnitude applied across the $_{MeO}$C$\cdots$C$_{OMe}$ coordinate (Fig. 2a, b and Supplementary Fig. 26). A reactive moiety coupled to an external force acting across a pair of its atoms is a widely used model in polymer mechanochemistry[30–33], suggesting that the observed variation in $\Delta G^{\ddagger}$ of isomerization across the series, including the allosteric effect of Pt binding, could be amenable to analysis within the formalism of mechanochemical kinetics[34].

The formalism of mechanochemical kinetics is designed to allow quantitative analysis and predictions of the effect of highly anisotropic molecular strain on reaction rates and selectivities when the strain results from interactions of intractably many molecular degrees of freedom[33]. This model was developed and validated for reactions of polymer chains stretched beyond their strain-free geometries by energy-dissipative environments such as elongational flow fields in rapidly flowing polymer solutions[35] or material under plastic deformations[36]. The measured kinetics of reactions in overstretched polymer chains[33] or non-macromolecular mimics reported to date[37] can be analyzed productively by representing the arbitrarily large reactant and its surroundings as the small reactive site coupled to an infinitely compliant harmonic stretching potential. Such a soft potential applies identical force on every conformer of the reactive site (and any transition state), independent of its molecular geometry. In this model, acceleration (or occasional inhibition[21]) of reactions in stretched reactants is primarily determined by the difference in the potential energy of the stretching potential coupled to the reactant and the rate-determining transition state, which is proportional to this force[38,39].

Under compressive load, the validity of the assumption that all conformers of a reacting molecule across all kinetically-significant states experience the same force is at best uncertain[16]. Within the standard mechanochemical model it is outright aphysical[34]. The reason is that the rate of a chemical reaction is sensitive to applied force only if the formation of the rate-determining transition state is accompanied by changes in the separation of the two atoms across which the force acts. A compressing potential soft enough for the

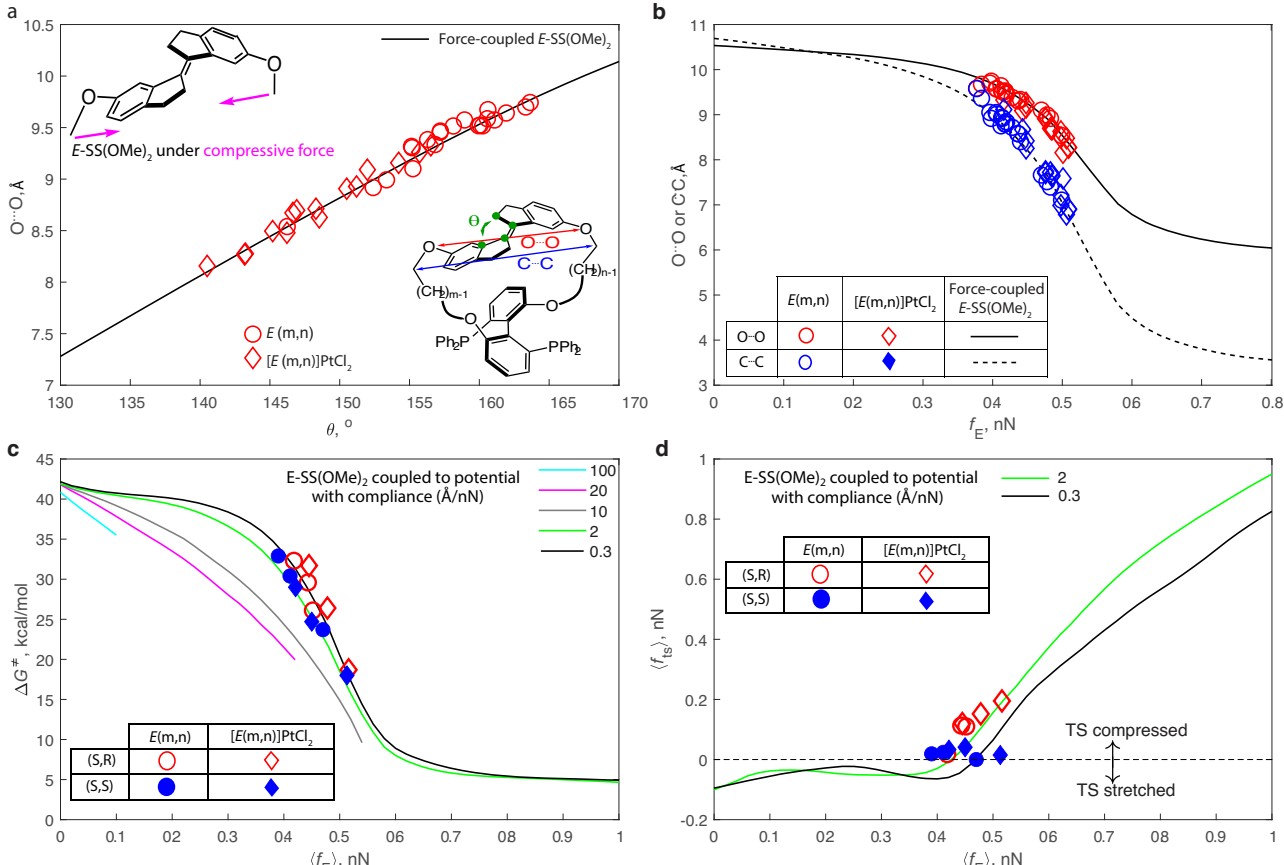

**Fig. 2 | Summary of the mechanochemical model of the measured isomerization kinetics. a**, **b** E-SS(OMe)$_2$ with compressive force acting on its methoxy C atoms reproduces the structures of the stiff stilbene moiety in the macrocycles. In (**a**) the underlying structural homology is illustrated by the correlation between the $_{Ar}$C-C = C-C$_{Ar}$ torsion and the O···O distance in all conformations of the E macrocycles following the same correlation in force-coupled SS(OMe)$_2$. In (**b**), the C···C and O···O distances in the conformers of the macrocycles are compared with the same distances in E-SS(OMe)$_2$ as a function of the applied force. **c** Calculated $\Delta G^{\ddagger}$ in the two diastereomeric series of the macrocycles are reproduced accurately using SS(OMe)$_2$ coupled to a compressive potential with compliance of 2 and 0.3 Å/nN respectively. Maximum $f_E$ that a compressive potential can apply decreases with increasing compliance, resulting in force-dependent activation free energy of isomerization, $\Delta G^{\ddagger}$, terminating below 1 nN for compliances >2 Å/nN. **d** Force exerted on SS(OMe)$_2$ by a compressive potential with a finite compliance decreases as the molecule progresses along the isomerization reaction path, resulting in lower ensemble-average force in the transition state, $\langle f_{ts} \rangle$, compared to that in the E isomer, $\langle f_E \rangle$. The plotted data is tabulated in Supplementary Tables 6–8.

applied force to remain approximately constant despite changes in the coupled molecular distance would require an equilibrium distance shorter than zero, which is obviously impossible.

Conversely, a physical compressive potential necessarily applies different force on each conformer of each kinetically-significant state. The stiffer this potential, the greater the variation of both the force among the conformers and the difference of the ensemble-average forces acting on the reactant and on the (rate-determining) transition state. As a result, the activation barrier of a reactant coupled to an external compressive potential depends both on its ensemble-average force and the potential compliance or, alternatively, the ensemble-average forces of the reactant and the rate-determining transition state.

To apply the formalism of mechanochemical kinetics to the macrocycles in this study, we first calculated $\Delta G^{\ddagger}$ of E→Z isomerization of SS(OMe)$_2$ coupled to harmonic potentials with compliances between 100 Å/nN (very soft) and 0.1 Å/nN (very stiff), solid lines in Fig. 2c. With its compliance fixed, the force applied by a potential on the coupled molecule is controlled by the equilibrium distance of the potential: the shorter the distance, the higher the force. At zero equilibrium distance this force reaches maximum, which increases as the potential stiffens (e.g., blue, magenta and grey lines). At the same ensemble-average force, $\langle f_E \rangle$, a soft potential reduces $\Delta G^{\ddagger}$ by more than

a stiffer potential because the change in its strain energy between the reactant and the transition state is proportional to the potential's compliance. As the latter decreases, its effect on the dependence of $\Delta G^{\ddagger}$ on the ensemble-average force experienced by E-SS, $\langle f_E \rangle$ decreases as well until the $\Delta G$ vs. $\langle f_E \rangle$ correlation becomes compliance-independent at -0.2 Å/nN.

To analyze mechanochemically the variation of the kinetic stability of E-stiff stilbene across both the metalated and unmetalled macrocycles, we first calculated the force on the methoxy C atoms of E-SS(OMe)$_2$, $f_E$, needed to reproduce the geometry of the corresponding moiety in each thermally accessible conformer of each E macrocycle. We then averaged resulting single-conformer $f_E$ in proportion to the Boltzmann weight of each corresponding conformer in the reactant ensembles. The correlation between these ensemble $\langle f_E \rangle$ and $\Delta G^{\ddagger}$ calculated for each macrocycle (symbols in Fig. 2c) follows closely the analogous correlations calculated for E-SS(OMe)$_2$ coupled to potentials with compliances of 2 Å/nN or 0.3 Å/nN, for the two diastereomeric series (Fig. 2c).

These calculations reveal that the exceptionally large allosteric kinetic effect of Pt binding to our macrocycles (up to 10$^4$-fold acceleration at 300 K) arises from only a small increase in the strain of the E-SS moiety, as evidenced by $\langle f_E \rangle$ of stiff stilbene in Pt-coordinated complexes exceeding that of the Pt-free precursors by only 30–80 pN

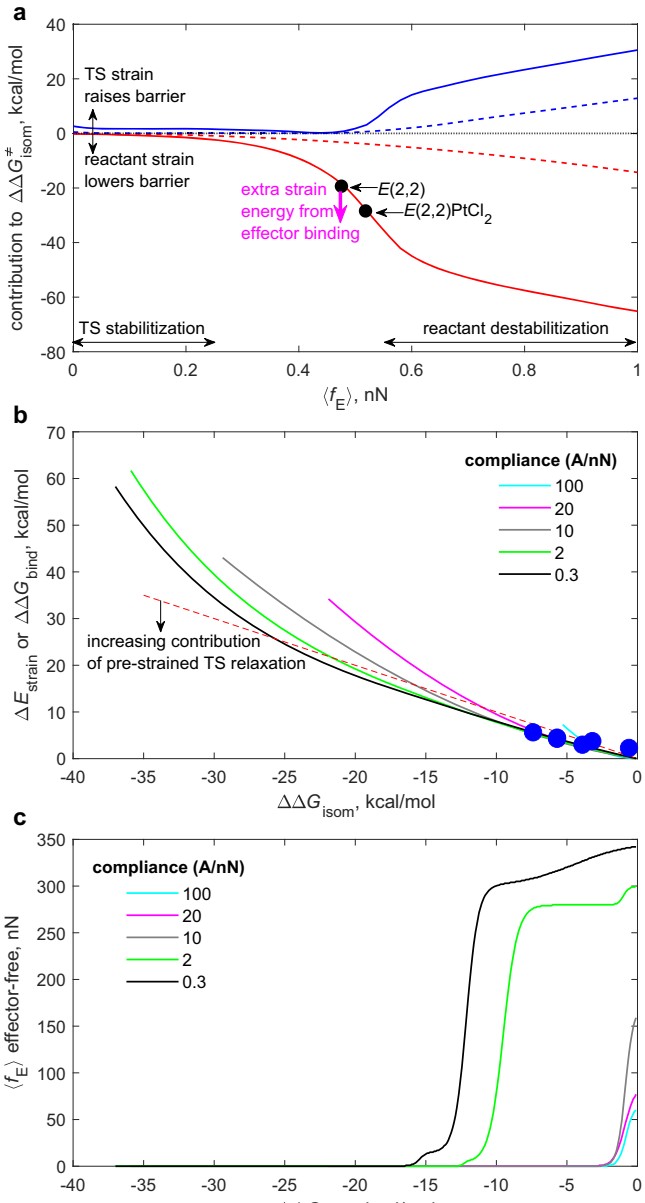

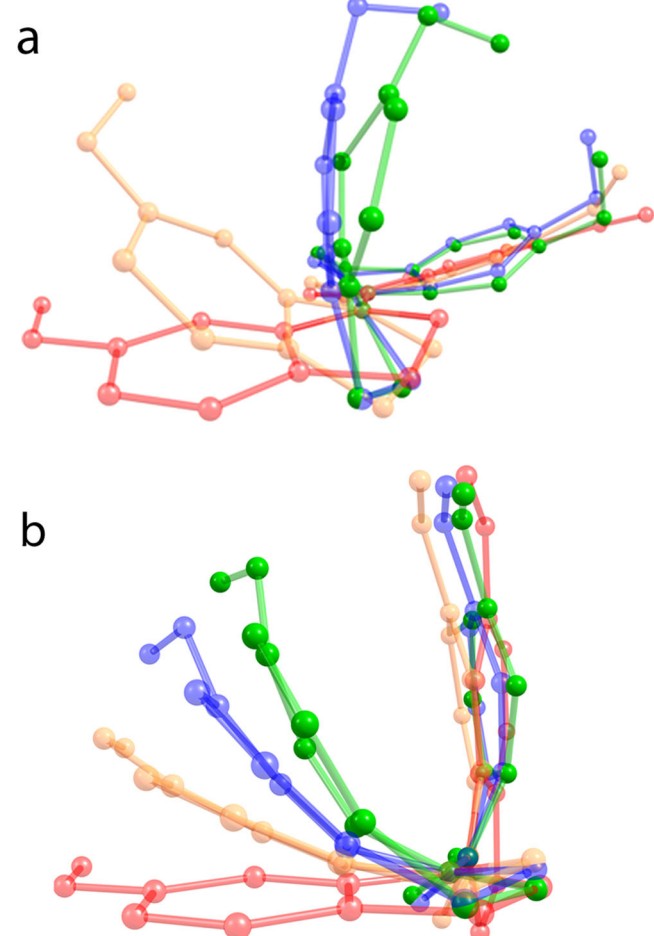

**Fig. 4 | The computed effect of compressive force on the geometry of stiff stilbene.** Overlay of the minimum-energy conformers of E-SS(OMe)$_2$ (**a**) and its transition state (**b**) in the absence of force (red) and coupled to compressive force of 0.5, 1 and 1.5 nN (light orange, blue and green, respectively). The view is along the C=C bond that isomerizes. Note the much greater bending of the indanone core in the E isomer compared to the TS at the same force.

**Fig. 3 | Mechanochemical analysis of allosteric acceleration. a** Strain energies of the substrate (solid lines) and the coupled compressive potential with compliance of 2 Å/nN (dashed lines) for the reactant (red) and the transition state (blue) as the function of the ensemble-average compressive force on E-SS, $\langle f_E \rangle$. **b** The minimum energy needed to reduce the isomerization barrier by $\Delta\Delta G^{\ddagger}_{isom} = \Delta G^{\ddagger}_{allost} - \Delta G^{\ddagger}_{free}$, where $\Delta G^{\ddagger}_X$ is the activation free energy of isomerization of free E-SS (X = free) or *E*-SS in an effector-bound allosteric substrate (e.g., macrocycles in Fig. 1, X = allost). For each $\Delta\Delta G^{\ddagger}_{isom}$, all combinations of the two $\langle f_E \rangle$ values corresponding to effector-free and effector-bound reactant were considered to identify the minimum necessary increase in the total reactant strain energy, $\Delta E_{strain}$. For macrocycles (blue dots), $\Delta\Delta G^{\ddagger}_{bind}$ describes the reaction free energy of PtCl$_2$ transfer between free and SS-coupled BiPHEP (eq. 1). **c** $\langle f_E \rangle$ of E-SS prior to effector binding giving the highest achievable $\Delta\Delta G^{\ddagger}_{isom}/\Delta E_{strain}$ ratio (see Supplementary Table 3 for the definition of all parameters of the mechanochemical model). The plotted data is tabulated in Supplementary Tables 7–9.

(7–15%). The mechanochemical formalism allows the underlying molecular basis to be understood and generalized by analyzing the $\Delta G^{\ddagger}$ vs. $\langle f_E \rangle$ correlations in Fig. 2c in terms of 4 components: the ensemble-average strain energies of the substrate (E-SS(OMe)$_2$ and the isomerization transition state) and those of the coupled constraining

potential, which represents the rest of the molecule, including the effector binding site, Fig. 3a.

The strain energies of both E-SS(OMe)$_2$ and its compressing potential increase monotonically with $\langle f_E \rangle$ (red curves). Conversely, those of the TS vary non-monotonically: they are non-zero at $\langle f_E \rangle = 0$ and decrease with increasing $\langle f_E \rangle$ up to 0.42 nN. This seemingly counterintuitive dependence is the direct consequence of the finite compliance of the potential. At $\langle f_E \rangle = 0$ the equilibrium length of the potential equals that of $_{MeO}C\cdots C_{OMe}$ in strain-free E-SS(OMe)$_2$, which exceeds the same internuclear distance of the strain-free transition state by ~3.8 Å (e.g., the red structures in Fig. 4). Consequently, a potential that applies no force on the E isomer ($\langle f_E \rangle = 0$), stretches the shorter TS (Fig. 2d). As the equilibrium distance of the constraining potential contracts with increasing $\langle f_E \rangle$, the corresponding $\langle f_{ts} \rangle$ decreases before becoming compressive. Above this switchover force, all 4 strain energies increase monotonically.

The approximately planar geometry of strain-free E-SS makes its $_{MeO}C\cdots C_{OMe}$ coordinate quite stiff at applied compressive force <0.3 nN, but the bending of the indanone core at higher force (Fig. 4a) causes a sharp contraction of both $_{MeO}C\cdots C_{OMe}$ and O$\cdots$O distances at $f_E = 0.35$–0.6 nN (Fig. 2b), accompanied by the steeply increasing molecular strain energy. Because the TS is stiffer and less anharmonic (Fig. 4b and Supplementary Fig. 27), its strain energy is less sensitive to

applied force, yielding a range of $\langle f_E \rangle$ (0.35–0.6 nN) where a small differential compression of the E isomer causes disproportionately large reduction in $\Delta G^{\ddagger}$. Note that because force acting on the transition state, $\langle f_{TS} \rangle$, cannot be directly experimentally controlled, but is uniquely defined by $\langle f_E \rangle$ and the compliance of the coupled potential, both of which are experimentally controllable, we find it more productive to discuss the properties of the transition state in terms of the force acting on the reactant ($\langle f_E \rangle$), rather than $\langle f_{TS} \rangle$.

This analysis suggests two limiting mechanisms of allosteric acceleration of a reaction. In one, both the reactant and the transition state in the absence of the effector are strained and effector binding does not alter the strain of the reactant but allows relaxation of the transition state. For SS this regime is accessible only at moderate substrate strain (e.g., $\langle f_E \rangle$<0.3 nN for E-SS(OMe)$_2$, the TS stabilization region in Fig. 3a). In the other limit, the effector-free substrate may be strained or unstrained but effector binding must increase the strain energy, $\Delta E_{strain}$, of the reactant considerably more than that of the transition state. In SS, this regime is only accessible for substrates that are already moderately strained in the absence of effector (the reactant destabilization in Fig. 3a), as they are in our macrocycles.

In allosteric reactants such $\Delta E_{strain}$ increases come at the expense of reduced binding affinity of the effector relative to free receptor (or any molecular architecture where receptor and substrate are conformationally uncoupled). For example, in our macrocycles, calculated $\Delta\Delta G^{\ddagger}_{bind}$ for transfer of PtCl$_2$ between E(n,m) and free BiPHEP (eq. 1) becomes increasingly unfavorable as the macrocycle shrinks (and hence gets more strained). Consequently, the highest practically achievable allosteric acceleration is likely limited by the acceptable reduction of the affinity of the coupled receptor for the chosen effector, and the attendant minimum necessary increase in the concentration of the effector. The ability to estimate the minimum effector-binding free energy, $\Delta\Delta G^{\ddagger}_{bind}$, that must be sacrificed to achieve desired acceleration of a substrate reaction is thus a primary determinant in the design of practical allosteric kinetic control.

$$(BiPHEP)PtCl_2 + E(n,m) \rightleftharpoons BiPHEP + E(n,m)PtCl_2 \qquad (1)$$

Figure 3b illustrates the relationship between $\Delta\Delta G^{\ddagger}_{bind}$, $\Delta E_{strain}$ and $\Delta\Delta G^{\ddagger}_{isom}$ for E stiff stilbene estimated by mechanochemical analysis. It plots the minimum effector-induced increase in the reactant strain energy, $\Delta E_{strain}$, needed to reduce the isomerization barrier by $\Delta\Delta G^{\ddagger}_{isom}$ below that of free E-SS(OMe)$_2$ for compressive potential of different compliances. The dashed red line corresponds to full $\Delta E_{strain}$ contributing to barrier lowering ($\Delta E_{strain} = -\Delta\Delta G^{\ddagger}_{isom}$). Values below this line reflect the contribution of relaxation of the pre-strained TS to $\Delta\Delta G^{\ddagger}_{isom}$. For E-SS, this contribution can exceed 50% for weak accelerations ($\Delta\Delta G^{\ddagger}_{isom} > -3$ kcal/mol), provided the substrate is significantly strained prior to effector binding (as quantified by its restoring force, e.g., $\langle f_E \rangle$ in Fig. 3c). At every $\Delta\Delta G^{\ddagger}_{isom}$, a stiffer potential requires a smaller effector-induced increase in the reactant strain energy, $\Delta E_{strain}$, than a softer one, which is opposite to what is calculated for tensile force[40]. This suggests that in general, increasing the stiffness of the receptor and the molecular segments connecting it to the substrate sites would reduce the loss of the effector affinity (i.e., smaller $\Delta\Delta G^{\ddagger}_{bind}$) to achieve a desired barrier lowering. One scenario in which a softer compressive potential (and hence the molecular segment it represents) may be advantageous is at low target barrier reductions (e.g., $\Delta\Delta G^{\ddagger}_{isom} > -3$ kcal/mol for E-SS(OMe)$_2$), where it decreases how much the substrate needs to be pre-strained to achieve the same allosteric coupling efficiency, i.e., the same $\Delta\Delta G^{\ddagger}_{isom}/\Delta\Delta G^{\ddagger}_{bind}$ ratio (e.g., grey vs. green curves in Fig. 3c).

Our macrocycles broadly follow these trends. For both diastereomers of E(2,2), (S,S)-E(2,3) and (S,R)-E(3,3), the $\Delta\Delta G^{\ddagger}_{isom}/\Delta\Delta G^{\ddagger}_{bind}$ ratio is within 90% of $\Delta\Delta G^{\ddagger}_{isom}/\Delta E_{strain}$ predicted by the mechanochemical model. In these macrocycles the reduction in the effector

affinity, $\Delta\Delta G^{\ddagger}_{bind}$, accounts for ~78% of observed barrier lowering, vs. 65–75% predicted by the model. These values confirm that the structure of these macrocycles provides a scaffold for remarkably efficient transmission of molecular strain across ~1 nm and ~100 non-H atoms and is amenable to both full atomistic description and simple quantitative analysis with a mechanochemical model.

The reported macrocycles represent one of the very few examples of synthetic molecules which exploit allostery to lower the activation barrier of a reaction below that in the equivalent non-allosteric reactant. The unusually large kinetic allosteric effects in these macrocycles result from the effector binding increasing the strain energy of the reactant without a concomitant increase in the strain energy of the corresponding transition state. The formalism of mechanochemical kinetics, generalized to compressive load, explains the observed trends quantitatively and suggests broad molecular-design approaches to achieving efficient allosteric acceleration of reactions. For example, the dependence of $\Delta G^{\ddagger}$ on force shown in Fig. 2c quantifies the importance of molecular compliance as a design feature to enhance or suppress the fraction of the effector binding energy that reduced the kinetic barrier. As a result, the allosteric sensitivity depends non-monotonically on applied force: higher force regimes do not necessarily lead to greater allosteric regulation. Consequently, the mechanochemical formalism may prove valuable for guiding the design of synthetic allosteric catalysts and for quantitative tests of molecular models of allosterically controlled enzymatic activity as resulting from structural transmission of molecular strain across suitably stiff portions of the biomolecular scaffolds[5]. In addition, the formalism applied here might inform approaches to improving the efficiencies of switching[23], energy storage[41] and work-generation[15] in increasingly diverse macrocyclic molecular switches[42–44] and machines[24,45–47] for a range of potential applications[48–50]. Equally important, E→Z isomerization of stiff stilbene reported here is only the 2nd reaction[17] whose kinetics has been demonstrated to be sensitive to compressive force. As such, it constitutes a valuable tractable model reaction on which to develop quantitative descriptions of mechanochemical kinetics across the full range of externally applied force.

## Methods
All reactions were performed under a nitrogen atmosphere in oven-dried glassware employing standard Schlenk or glovebox techniques unless noted otherwise. Nitrogen-flushed plastic syringes and oven-dried stainless steel cannulas were employed for reagent transfer. NMR spectra were obtained at 25 °C unless noted otherwise. $^{31}$P NMR spectra were referenced using absolute frequency referencing in Mnova software or from trimethylphosphine oxide internal standard. NMR probe temperature were determined from a single scan of neat ethylene glycol (high temperature) or neat methanol (low temperature) with accuracy of ±1 °C[51]. Anhydrous solvents were obtained either from Sigma-Aldrich in Sure/Seal™ containers or were dried and degassed using an Innovative Technologies PureSolv solvent purification system. All deuterated solvents were obtained from Cambridge Isotope Laboratory and were dried over activated 3 Å molecular sieves. Freshly opened anhydrous p-xylene-$d_{10}$ was degassed via three freeze-pump-thaw cycles and stored in a glovebox. All other reagents were purchased from major chemical suppliers and were used as received. All macrocycles possess an S-configuration about the MeOBiphep axis. Error values associated with individual rate constants refer to the standard deviation of the linear regression.

### Synthesis of (P–P)PtCl$_2$ complexes
**[(S,S)-E(3,3)]PtCl$_2$**[19]. A solution of (COD)PtCl$_2$ (26.3 mg, $7.0 \times 10^{-2}$ mmol) in CD$_2$Cl$_2$ (0.2 mL) was added dropwise via syringe to a solution of (S,S)-E(3,3) (63.1 mg, $7.0 \times 10^{-2}$ mmol) in CD$_2$Cl$_2$ (0.5 mL) in an NMR tube, which was sealed under nitrogen. The reaction mixture was monitored by $^{31}$P NMR spectroscopy to achieve ≥95%

conversion (1 h). The solution was diluted with diethyl ether (20 mL) and the resulting precipitate was filtered and washed with diethyl ether. Recrystallization from a saturated dichloromethane solution layered with diethyl ether at −20 °C gave pure [(S,S)-E(3,3)]PtCl$_2$ (69 mg, 82%) as colorless crystals. $^1$H NMR (500 MHz, CD$_2$Cl$_2$): δ 7.78 (s, br, 4H), 7.66–7.59 (m, 4H), 7.50–7.38 (m, 8H), 7.25 (t, J = 7.5 Hz, 4H), 7.18 (d, J = 8.1 Hz, 2H), 6.95 (s, 2H), 6.88 (td, J = 8.3, 2.3 Hz, 2H), 6.75–6.63 (m, 4H), 6.26 (d, J = 8.4 Hz, 2H), 4.31–4.19 (m, 4H), 3.38–3.28 (m, 2H), 3.17–2.80 (m, 10H), 1.65–1.49 (m, 4H). $^{13}$C{$^1$H} NMR (126 MHz, CD$_2$Cl$_2$): δ 156.9 (d, J = 5.4 Hz), 156.9 (d, J = 5.4 Hz), 155.4, 144.9, 141.2, 135 2 (d, J = 5.4 Hz), 135.1 (d, J = 4.7 Hz), 134.7, 131.6, 130.7, 129.1 (d, J = 4.4 Hz), 129.0 (d, J = 6.0 Hz), 128.9 (d, J = 6.0 Hz), 128.7 (d, J = 5.2 Hz), 128.6 (d, J = 4.4 Hz), 128.1 (d, J = 5.0 Hz), 127.2 (d, J = 5.7 Hz), 127.1 (d, J = 5.7 Hz), 126.4 (d, J = 7.6 Hz), 126.3 (d, J = 7.6 Hz), 126.0, 124.8, 124.7 (d, J = 4.3 Hz), 124.0 (d, J = 5.5 Hz), 123.5 (d, J = 5.5 Hz), 118.2, 113.0, 110.7, 65.7, 65.0, 36.1, 31.8, 27.4. $^{31}$P NMR (162 MHz, CD$_2$Cl$_2$) δ 8.78 (s, J$_{PtP}$ = 3664 Hz). HRMS-ESI (m/z) calcd (found) for C$_{60}$H$_{52}$ClO$_4$P$_2$Pt [M−Cl]$^+$: 1129.2681 (1129.2685). Complexes [(R,S)-E(3,3)]PtCl$_2$ and [(R,S)-E(2,3)]PtCl$_2$ were synthesized from (S,R)E(3,3) and (S,R)-E(2,3), respectively, employing analogous procedures.

**[(S,R)-E(3,3)]PtCl$_2$[19]**. Pale yellow crystals, 58%. $^1$H NMR (500 MHz, CD$_2$Cl$_2$): δ 7.70 (br s, 4H), 7.51 (m, 4H), 7.38–7.29 (m, 8 H), 7.17 (t, J = 7.0 Hz, 4 H), 7.13 (d, J = 8.0 Hz, 2H), 7.03 (d, J = 1.5 Hz, 2H), 6.69–6.67 (m, 4 H), 6.56 (dd, J = 10.5, 8.0 Hz, 2H), 5.70 (d, J = 8.5 Hz, 2H), 4.29 (ddd, J = 12, 6.5, 4.5 Hz, 2H), 3.99 (ddd, J = 12.5, 7.0, 5.0 Hz, 2H), 3.20 (td, J = 11.5, 5.0 Hz, 2 H), 3.00-2.75 (m, 10 H), 1.76 (m, 2 H), 1.61 (m, 2 H). $^{13}$C{$^1$H} NMR (126 MHz, CD$_2$Cl$_2$): δ 157.0 (d, J = 5.4 Hz), 156.9 (d, J = 5.4 Hz), 156.4, 144.7, 141.9, 135.1 (d, J = 6.0 Hz), 135.0 (d, J = 6.0 Hz), 134.9, 131.5, 130.6, 129.1 (d, J = 6.0), 128.9 (d, J = 8.0 Hz), 128.8 (d, J = 8.0 Hz), 128.7 (d, J = 7.0 Hz), 128.5 (d, J = 5.7 Hz), 128.2 (d, J = 7.4 Hz), 128.1 (d, J = 7.4 Hz), 127.2 (d, J = 7.4 Hz), 127.1, (d, J = 7.4 Hz), 126.4 (d, J = 9.4 Hz), 126.3 (d, J = 9.4 Hz), 125.9, 124.8 (d, J = 5.8 Hz), 124.7 (d, J = 5.8 Hz), 124.4 (d, J = 7.4 Hz) 123.9 (d, J = 7.8 Hz), 118.7, 113.0, 111.7, 67.0, 66.7, 35.7, 31.6, 30.6. $^{31}$P NMR (162 MHz, CD$_2$Cl$_2$): δ 8.79 (s, J$_{PtP}$ = 3652 Hz). HRMS-ESI (m/z) calcd (found) for C$_{60}$H$_{52}$ClO$_4$P$_2$Pt [M−Cl]$^+$: 1129.2681 (1129.2672).

**[(S,R)-E(2,3)]PtCl$_2$[19]**. White solid, 69%. $^1$H NMR (500 MHz, CD$_2$Cl$_2$): δ 7.84 (qd, J = 13.4, 7.8 Hz, 4H), 7.63 (ddd, J = 31.8, 11.8, 7.4 Hz, 5H), 7.54–7.30 (m, 13H), 7.19 (dt, J = 17.0, 7.8 Hz, 4H), 7.00 (s, 1H), 6.86 (d, J = 8.5 Hz, 3H), 6.69 (ddq, J = 30.5, 21.8, 12.3, 9.9 Hz, 3H), 6.20 (d, J = 8.6 Hz, 1H), 5.48 (d, J = 8.4 Hz, 1H), 4.34 (dd, J = 12.2, 6.1 Hz, 2H), 4.04 (dt, J = 42.9, 10.6 Hz, 3H), 3.21–2.76 (m, 10H), 2.63 (q, J = 10.7 Hz, 1H), 2.40 (t, J = 10.9 Hz, 1H), 2.12 (td, J = 11.2, 6.0 Hz, 1H), 1.69 (dq, J = 13.7, 7.4 Hz, 1H), 1.42 (dt, J = 16.9, 9.4 Hz, 1H). $^{13}$C{$^1$H} NMR (126 MHz, CD$_2$Cl$_2$): δ 158.09, 157.99, 157.17, 157.11, 157.01, 155.59, 146.68, 145.81, 144.09, 142.02, 135.77, 135.68, 135.58, 135.52, 135.44, 135.37, 135.29, 134.90, 132.27, 132.26, 131.95, 131.94, 131.05, 131.03, 130.99, 130.98, 129.29, 129.25, 129.19, 129.02, 128.82, 128.74, 128.51, 128.27, 128.18, 128.14, 128.05, 127.61, 127.58, 127.52, 127.49, 126.81, 126.12, 125.43, 125.36, 124.90, 124.83, 124.48, 124.44, 123.99, 123.94, 120.38, 118.90, 117.99, 113.80, 113.09, 109.34, 71.13, 68.47, 66.21, 63.60, 37.32, 36.07, 32.39, 32.14, 30.97. $^{31}$P NMR (202 MHz, CD$_2$Cl$_2$): δ 9.44, 8.44 (ABq, J$_{PP}$ = 18 Hz, J$_{PtP}$ = 3670, 3662 Hz). HRMS-ESI (m/z) calcd (found) for C$_{59}$H$_{50}$ClO$_4$P$_2$Pt [M−Cl]$^+$: 1115.2524 (1115.2508).

**[(S,R)-E(2,2)]PtCl$_2$[19]**. A solution of (NBD)PtCl$_2$ (6.2 mg, 1.7 × 10$^{-2}$ mmol) in DMF-$d_7$ (0.3 mL) was added dropwise via syringe to a septum-capped NMR tube containing a solution of (S,R)-E(2,2) (15.9 mg, 1.8 × 10$^{-2}$ mmol) in DMF-$d_7$ (0.3 mL) cooled at −78 °C. The contents of the tube were mixed thoroughly at −78 °C and placed in the probe of an NMR spectrometer pre-cooled at −30 °C. $^{31}$P NMR analysis after 10 min revealed quantitative formation of [(S,R)-E(2,2)]PtCl$_2$ (9.37 ppm, s). Thermally unstable [(S,R)-E(2,2)]PtCl$_2$ was characterized in solution without isolation. $^1$H NMR (700 MHz, DMF-$d_7$, −30 °C): 7.64 (m, 6 H), 7.52 (d, J = 7.0 Hz, 2 H), 7.50 (t, J = 7 Hz, 8H), 7.39 (d, J = 8.4 Hz, 2 H), 7.01 (t, J = 7.7 Hz, 2H), 6.98 (d, J = 7.7 Hz, 2H), 6.78 (d, J = 1.5 Hz, 2H), 6.89 (d, J = 8.4 Hz, 2H), 6.59 (t, J = 8.4, Hz, 2H), 4.13 (m, 4 H), 3.80 (m, 2 H), 3.25 (m, 2 H), 3.01 (m 4 H), 2.86 (dd, J = 7.0, 13 Hz, 2 H), 2.81 (dd, J = 5.6, 12 Hz, 2 H). $^{31}$P NMR (283 MHz, CD$_2$Cl$_2$): δ 9.37 (J$_{PtP}$ = 3650 Hz).

**Isomerization of [(S,R)-E(2,3)]PtCl$_2$[18]**. A solution of [(S,R)-E(2,3)]PtCl$_2$ (8.6 mM) containing free (S,R)-E(2,3) (~8.6 mM) was generated via addition of DMF-$d_7$ (0.60 mL) via syringe to a septum-capped NMR tube containing (NBD)PtCl$_2$ (1.85 mg, 5.17 × 10$^{-3}$ mmol, 8.6 mM) and (S,R)-E(2,3) (9.12 mg, 1.03 × 10$^{-2}$ mmol) at 25 °C. The tube was shaken until the mixture was homogeneous, inserted into the probe of an NMR spectrometer preheated at 80 °C, and analyzed periodically by $^{31}$P NMR spectroscopy. The concentration of [(S,R)-E(2,3)]PtCl$_2$ was determined by integrating the resonances corresponding to [(S,R)-E(2,3)]PtCl$_2$ at δ 6.39 and 5.39 (ABq, J$_{PP}$ = 19 Hz) and [Z(2,3)]PtCl$_2$ at δ 4.94 (s) assuming quantitative E to Z isomerization; this assumption was supported by the absence of any additional resonances in the $^{31}$P NMR spectrum throughout complete conversion of [(S,R)-E(2,3)]PtCl$_2$ to [Z(2,3)]PtCl$_2$. Furthermore, upon complete (≥95%) isomerization of [(S,R)-E(2,3)]PtCl$_2$ to [Z(2,3)]PtCl$_2$, no significant isomerization of free (S,R)-E(2,3) was observed, as judged by $^{31}$P NMR spectroscopy. The first-order rate constant for the conversion of [(S,R)-E(2,3)]PtCl$_2$ to [Z(2,3)]PtCl$_2$ was obtained from a linear plot of ln{[(S,R)-E(2,3)]PtCl$_2$}[17] versus time through 3 half-lives (Supplementary Table 1, entry 1). First-order rate constants for the conversion of [(S,R)-E(2,3)]PtCl$_2$ to [Z(2,3)] PtCl$_2$ were determined as a function of temperature from 64 to 80 °C employing similar procedures (Supplementary Table 1, entries 1–5). A plot of ln(k/T) versus reciprocal temperature provided the activation parameters for isomerization (Supplementary Table 2).

**Isomerization of [(S,S)-E(3,3)]PtCl$_2$ and [(S,R)-E(3,3)]PtCl$_2$**. First-order rate constants for the isomerization [(S,S)-E(3,3)]PtCl$_2$ at 125 °C in DMF-$d_7$ (Supplementary Table 1, entry 6) and for the isomerization of [(S,R)-E(3,3)]PtCl$_2$ at 121 °C in DMF-$d_7$ (Supplementary Table 1, entry 7) were determined employing procedures similar to those used to determine the first-order rate constants for the isomerization of [E(2,3)]PtCl$_2$. Both [(S,S)-E(3,3)]PtCl$_2$ and [(S,R)-E(3,3)]PtCl$_2$ isomerize to form [Z(3,3)]PtCl$_2$[19].

**Isomerization of [(S,R)-E(2,2)]PtCl$_2$**. A solution of [(S,R)-E(2,2)]PtCl$_2$ containing free (S,R)-E(2,2) was generated by addition of a solution of (NBD)PtCl$_2$ (1.85 mg, 5.17 × 10$^{-3}$ mmol) in DMF-$d_7$ (0.60 mL) via syringe to a septum-capped NMR tube containing (S,R)-E(2,2) (8.97 mg, 1.03 × 10$^{-2}$ mmol) at −30 °C. The contents of the tube were mixed thoroughly at −30 °C, the tube was placed in the probe of an NMR spectrometer pre-cooled at 6 °C, at which time the $^{31}$P NMR spectrum revealed quantitative formation of [(S,R)-E(2,2)]PtCl$_2$. The solution was analyzed periodically by $^{31}$P NMR spectrometry by integrating the resonances corresponding to [(S,R)-E(2,2)]PtCl$_2$ (δ 6.27, s) and [Z(2,2)] PtCl$_2$ (δ 5.09, s). First-order rate constants (Supplementary Table 1, entries 8–12) and activation parameters (Supplementary Table 2) for the isomerization [(S,R)-E(2,2)]PtCl$_2$ to [Z(2,2)]PtCl$_2$ were determined employing procedures similar to those used to determine the first-order rate constants for the isomerization of [(S,R)-E(2,3)]PtCl$_2$.

**Isomerization of (S,R)-E(2,2)**. A septum-capped NMR tube containing a solution (S,R)-E(2,2) (4.4 mg, 5.17 × 10$^{-3}$ mmol, 8.61 mM) in p-xylene-$d_{10}$ (0.6 mL) was placed in the probe of an NMR spectrometer pre-heated at 84 °C and analyzed periodically by $^{31}$P NMR spectroscopy. The concentration of (S,R)-E(2,2) was determined by integrating the resonances corresponding to E(2,2) at δ −7.04 (s) and Z(2,2) at δ −9.34 (s) assuming quantitative E to Z isomerization; this assumption was

supported by the absence of any additional resonances in the $^{31}$P NMR spectrum throughout complete conversion of (S,R)-E(2,2) to Z(2,2). The first-order rate constant for the conversion of (S,R)-E(2,2) to Z(2,2) was obtained from the linear plot of ln[(S,R)-E(2,2)] versus time through 3 half-lives (Supplementary Table 1, entry 13). First-order rate constants for the conversion of (S,R)-E(2,2) to Z(2,2) were determined as a function of temperature from 69 to 84 °C employing a similar procedure (Supplementary Table 1, entries 13–16). A plot of ln(k/T) versus reciprocal temperature provided the activation parameters for isomerization (Supplementary Table 2).

**Isomerization of (S,R)-E(2,3), (S,S)-E(3,3), and (S,R)-E(3,3).** First-order rate constants and activation parameters for the isomerization of (S,R)-E(2,3) were obtained employing procedures similar to that used to determine the first-order rate constants and activation parameters for the isomerization of (S,R)-E(2,2) (Supplementary Table 1, entries 17–22; Supplementary Table 2). The first-order rate constant for the isomerization of (S,R)-E(3,3) in p-xylene-$d_{10}$ Supplementary Table 1, entry 23) at 126 °C was obtained employing a procedure similar to that used to determine the first-order rate constants for the isomerization of (S,R)-E(2,2). The first-order rate constant for the isomerization of (S,S)-E(3,3) was obtained by heating an NMR tube containing a solution of (S,S)-E(3,3) in p-xylene-$d_{10}$ in an oil bath at 131 °C, which was removed periodically, cooled immediately to 0 °C, and analyzed by $^{31}$P NMR spectroscopy at 25 °C (Supplementary Table 1, entry 24).

**Solvent effect on ligand isomerization.** Separate solutions of (S,R)-E(2,3) (8.6 mM) in DMF-$d_7$ (0.60 mL) and p-xylene-$d_{10}$ were heated together at 120 °C, removed periodically, cooled immediately to 0 °C, and analyzed at identical time points by $^{31}$P NMR spectroscopy at 25 °C. The first-order rate constant for the isomerization of (S,R)-E(2,3) in DMF-$d_7$ ($k_{obs} = 9.8 \pm 0.3 \times 10^{-5}$ s$^{-1}$) and p-xylene-$d_{10}$ ($k_{obs} = 9.3 \pm 0.3 \times 10^{-5}$ s$^{-1}$; Table 1, entry 22, Fig. S25) differed by ~5%.

### DFT calculations
All calculations were performed with Gaussian 16.C in vacuum; the Berny algorithm was applied to locate stationary points and tight convergence criteria and ultrafine integration grids were used in optimizations and frequency calculations. Reactant conformers of the macrocycles were generated and optimized as previously described[18], except for the used model chemistry (B3LYP/def2SVP); we previously demonstrated that the def2SVP basis set leads to faster convergence of geometry optimizations of complexes of heavy transition metals and yields kinetic barriers for reactions of such complexes in better agreement with experiment than LANL2DZ[52]. Free energies of individual conformers were sums of the single-point electronic energy calculated at the (u)BMK/def2SVP level and the corresponding thermodynamic corrections calculated statistically-mechanically with all frequencies <500 cm$^{-1}$ replaced with 500 cm$^{-1}$ to avoid artifactually large contributions of vibrations to $\Delta G^{\ddagger}$[53]. Further details are provided in the Supplementary Information.

### The mechanochemical model of allosteric isomerization
For each conformer of E-SS(OMe)$_2$ and TS-SS(OMe)$_2$ we calculated the equilibrium distance, $q_{sp}^o$, of every compressive potential with compliance, λ, from 100 Å/nN to 0.1 Å/nN in 0.1 Å/nN increments needed to exert force, f, from −0.2 nN to 1 nN, or the highest compressive force at which the conformer exists, in 1 pN increments (f > 0 corresponds to compression) according to formula, $q_{sp}^o = q_i(f) - f\lambda$, where $q_i(f)$ is the $_{MeO}$C$\cdots$C$_{OMe}$ distance of conformer i of SS(OMe)$_2$ in equilibrium with force f, derived quantum-chemically (see the preceding section). Linear interpolation of this data yielded the values of relative free energy, $_{MeO}$C$\cdots$C$_{OMe}$ and its restoring force of each conformer of SS(OMe)$_2$ for each combination of λ and $q_{sp}^o$ (primary correlation matrix). All conformers coupled to the potential of the same λ and $q_{sp}^o$ comprise a

conformational ensemble, characterized by ensemble-average parameters, such as force, molecular strain energy and the potential energy of the constraint. The activation free energy of SS isomerization coupled to a potential with given λ and $q_{sp}^o$ is derived from the corresponding E and TS ensembles (Supplementary Tables 7 and 8). A subset of this data is plotted in Figs. 2c, d and 3a.

Whereas isomerization $\Delta G^{\ddagger}$ of SS(OMe)$_2$ is uniquely defined by specifying λ and $q_{sp}^o$, the change in $\Delta G^{\ddagger}$ caused by a given change in $q_{sp}^o$ (which represents the contraction of the BIPHEP containing strap upon PtCl$_2$ binding in our experiments and more generally, the mechanism by with effector binding lowers the kinetic barrier), $\Delta\Delta G^{\ddagger}_{isom}$, (Fig. 3b, c) is generally not. Consequently, to identify the highest achievable efficiency of allosteric control for potential of each compliance, we calculated by interpolation of the master correlation matrix all $\Delta q_{sp}^o$ values needed to achieve each $\Delta\Delta G^{\ddagger}$ value from 0 to −37 kcal/mol in 0.1 kcal/mol increments and for each $\Delta\Delta G^{\ddagger}$ selected the combination of two $q_{sp}^o$ (before and after binding) that corresponding to the smallest increase in the energy of the potential (Supplementary Table 9). This data is plotted in Fig. 3b, c.

### Data availability
The characterization and kinetic data generated in this study, and Cartesian coordinates of the converged geometries of the minimum-energy conformers of all kinetically significant states have been deposited in the Duke Research Data Repository under Creative Commons CC0 1.0 Universal license [https://doi.org/10.7924/r4474k10q]. The summary characterization and kinetic data generated in this study are provided in the Supplementary Information/Source Data file. All data is available from the corresponding authors upon request.

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

## Acknowledgements

This material is based upon work supported by the U.S. Department of Energy, Office of Science, Office of Basic Energy Sciences program under Award Number DE-SC0018188 (R.A.W. and S.L.C.) and used the Extreme Science and Engineering Discovery Environment (XSEDE), which is supported by National Science Foundation grant number ACI-1548562, with computational resources provided by the SDSC under allocation TG-CHE140039. Acknowledgment is made to the Donors of the American Chemical Society Petroleum Research Fund for partial support of this research under grant 58885-ND7 (to R.B.). Computations reported here relied on work partially supported by the Engineering and Physical Sciences Research Council under grant EP/L000075/1 (to R.B.) and National Science Foundation grant number CHE-1808518 (to S.L.C.). We thank Dr. Benjamin Bobay (Duke University) for assistance with NMR spectroscopy and Daniel Duan (Duke University) for performing additional characterization experiments.

## Author contributions

Y.Y. performed the synthesis and experiments. R.T.O. performed the calculations. R.B. supervised the computational work. R.A.W. and S.L.C. supervised the experimental work. The manuscript was written through contributions of all authors. All authors have given approval to the final version of the manuscript.

## Competing interests

The authors declare no competing interests.
