## [Peer Review File · Nature Communications]

Allosteric control of olefin isomerization kinetics via remote metal binding and its mechanochemical analysisReviewers' Comments:

Reviewer #1:

Remarks to the Author:

The manuscript presented by Craig and coworkers reports an up to 10000-fold thermal E-to-Z isomerization rate when a metal binding on the remote receptor of a molecular probe (stiff stilbene). Moreover, they perform DFT calculations to simulate the strain transportation process and discuss the relation between distance, dihedral angle, force, compliance, and energy difference (strain energy, isomerization energy, and transition energy barrier). Based on our research on related systems from the research team's prior publication record, there are three closely related papers using such stiff stilbene diphosphine system: 1) in Pd-catalyzed C-C bond coupling reactions (Angew 2014); 2) in Pt-induced reductive elimination (Chem Sci 2021) and 3) in Pd-induced oxidative addition (JACS 2020).

In this manuscript, the authors evaluate the mechanochemical efficiency of increasing strain energy to lower isomerization energy and the influence of binding affinity. The assumption is supported by kinetic studies and DFT calculations. We have to admit that the simulation design is well-thought and well-executed in this work. However, the same system has been studied in an author's previous paper Chem. Sci., 2021, 12, 11130 (doi: 10.1039/d1sc03182a). In that paper, they discuss the relationship between the mechanical force and the Pt complex's reductive elimination rate, where the force applied/stiff stilbene conformation relationship was well-established by both experimental and theoretical approaches. To some extent, this manuscript looks very similar to that (reversed story, how compressive force influences the E-Z isomerization in this paper vs. how E-Z isomerization affects the compressive force imposed on diphosphine ligand and thus on the rate of reductive elimination). And the conclusion or message of this paper is interesting but not general. It's hard to foresee how other scientists can build on this work. Thus in my view, this manuscript is not sufficiently novel and general for publication in Nature Communications.

Here are some comments to help the author improve their manuscript:

1. A general scheme illustrating how the chemistry reported here to other general phenomena or science problems would help the reader to grasp the essence authors wanted to express.
2. In Figure 1, The author shows Z(m,n) can isomerization to E(m,n) under 365 nm light radiation. However, once forming [Z(m,n)]PtCl₂, will the strain coming from the metal center prevents the stiff stilbene from isomerizing under the current radiation condition?
3. The calculation in this paper is fruitful. However, there are too many physical quantities in this paper (>10), which makes the paper confusing and difficult to follow. I recommend listing all the quantities in the ESI and specifying their meaning, how they are defined, and calculation methods. This would definitely help readers to closely follow the discussion of the change of all the parameters.
4. It seems like the calculation energy fits well with the experiment results, while the calculation level (u)BMK/def2SVP//(u)B3LYP/def2SVP (line 90) is not good enough for calculations in this system. There are mainly three questions: a) Why change functional B3LYP to an unpopular functional BMK? Has the author done any benchmarks about different functionals in your system? b) It is strange that the optimization of such organometallic compounds using B3LYP without empirical dispersion, have the authors attempted to add the empirical dispersion to improve the structural optimization accuracy (and it's nearly free for calculation)? c) For a system in this scale (~100+ atoms), the single-point calculation level is commonly upgraded to 3- ζ (e.g. def2-TZVP for Ahlrichs basis sets). The author might need to recalculate the single-point energy to see whether more accurate energy they can achieve.
5. The calculation data is not fully provided. The coordinates and parameters (e.g. energy, force...) of binding affinity, mechanochemistry investigations, and transition state calculation should be

supplemented.

6. The term "reactor, receptor, spring, and effector" is confusing in this system. Thus, A figure showing the position of these components in the molecule is recommended to draw.

7. line 152 "... compliance-independent at $\sim 0.2 \text{ \AA/nm}$." Does the author mean " $\sim 0.2 \text{ \AA/nN}$ "?

8. line 191 "...yielding a range of $\langle fE \rangle$ (0.35-0.6 nN) ." Is the value the same as line 189, "...both MeOC-COMe and O-O distances at fE of 0.35 – 0.6 nN (Fig. 2b)" and fE should be fTS.

9. Figure S26 in ESI, could the author add legends on these figures?

10. Labels a and b are missing in Figure 4.

11. The manuscript is not an easy read in general. One possible reason is that the authors used mainly very long sentences throughout. A bit of proofreading and a touch on the sentence structure would help.

Reviewer #2:

Remarks to the Author:

In their manuscript "Allosteric control of olefin isomerization kinetics via remote metal binding and its mechanochemical analysis" Yu et al. describe how Pt coordination to a (S)-BIPHEP unit in a photoswitchable macrocycle affects the thermal E->Z double bond isomerization of a stiff stilbene unit.

In the first section of the manuscript, the authors provide experimental data for the isomerization kinetics of macrocycles with three different ring-sizes (E(2,2), E(2,3), E(3,3)) in their complexed and un-complexed form. Generally speaking, the activation free energies increase with increasing ring-size and decrease upon metal coordination. In the second section, an in-depth mechanochemical model for the observed allosteric acceleration of the thermal olefin isomerization is discussed. The theoretical model implies that a compressive force acting on the stiff stilbene unit is causing the rate acceleration of the reaction. The experimental and computational methods applied in this study seem sound.

In my opinion, a detailed mechanochemical description for the influence of a compressive force on the reactivity of a molecular entity is interesting and the fact that a small increase in strain accelerates thermal E->Z olefin isomerization dramatically is remarkable. The insights this study provides are important for various areas in chemistry and materials science.

While I think that the authors found an elegant system to describe the effect of compressive force on the double bond isomerization of stiff stilbene, I still have a few comments that should be addressed before publication:

- The readership of Nature Communications is very broad and some parts of the manuscript are a bit difficult to follow. It would be nice to show the general concept of the paper in figure 1 in a simplified manner. A schematic representation how the coordination of Pt applies a compressive load on the stiff stilbene unit would, in my opinion, improve the comprehensibility of the paper a lot.

- The authors mainly discuss how metal coordination accelerates olefin isomerization by imposing a compressive load on stiff stilbene. A similar (and intuitive) mechanochemical effect seems to apply by changing the size of the macrocycle from (2,2) to (2,3) to (3,3). This should be discussed by the authors in more detail in the main text.

- Related to the comment above, macrocyclic molecular switches (such as hydrazones, see e.g.

10.1021/jacs.8b07612 and 10.1021/jacs.2c05384) and machines (such as molecular motors, see e.g. 10.1002/anie.202104285, 10.1038/s41565-021-01021-z, 10.1021/jacs.2c02547 and reference 23) attracted quite some interested in the last couple of years. Understanding how macrocycle size affects the isomerization behavior of such systems from a mechanochemical point of view would be of great interest for these areas of research. By connecting their study to (at least) the studies above (the list of examples is by no means exhaustive), the authors would put their work in a broader scientific context and could diversify the relatively small number of references provided in the current version of the manuscript.

- The thermal E->Z isomerization was followed by ³¹P NMR spectroscopy and the rate plots are provided in the SI but not the corresponding NMR spectra. In order to better evaluate the kinetic experiments, the authors should provide the respective ³¹P NMR spectra (either superimposed or stacked) as well.

- To be more consistent and avoid confusion, the authors should indicate which diastereomer was formed for E(2,2) and E(2,3) and use the appropriate nomenclature throughout the manuscript.

Minor comments:

- p2, line 43 and 45: Photolysis should be changed to photoisomerization
- p3, line 75: Furthered should be changed to further
- p7, figure 4: The caption refers to a) and b), which are not shown in the figure

In conclusion, I think that the study is of high interest for the broad scientific readership of this journal and I therefore recommend publication in Nature Communications after the suggested changes.

Reviewer #3:

Remarks to the Author:

The work presented by Yu et al is a clear example of mechanochemistry as an efficient tool to modulate the thermal reactivity of some molecules, in this case the isomerization of stiff-stilbene. This study is the follow-up of a previous work of this research group, from a more computational and theoretical point of view. The used methods are explained in detail and can be applied to other systems. The article is well-written and all the conclusions are well supported by the results. The findings and rationalization show the key role of allostery, motivating its use as a tool to enhance the efficiency of some thermal reactions. For all these reasons, I recommend the publication of this work after some minor revisions.

1) In lines 43 and 45. The E isomers are not form by photolysis of the Z isomers, as no bonds are broken in this process. It is a photoisomerization completed after irradiation of the Z isomer.

2) In the footnote of Table 1: b) is missing.

3) Apart from rationalizing the effect of Pt coordination on the free-energies (table 1) it can also be discussed the differences between the (2,2), (2,3) and (3,3) macrocycles. For E(2,2) is the lowest one while the values of (2,3) and (3,3) are similar. Which is the reason behind this trend?

4) Along the manuscript, please unify the nomenclature of E-stiff (or E stiff). Also, E and Z should be written in italic.

5) Why in Table S2 is missing the data for macrocycle (3,3)?

6) In line 161, when rationalizing the larger strain when Pt-coordinated. Could it be possible to

compare C=C bond length for the reactant and TS structures of Pt-free and Pt-coordinated macrocycles (or different strengths of the applied force) to check if the isomerizable double bond is weakened (similar to Figure S26)? In the SI is given the analysis of the torsion for different force magnitudes for the TS structure. A similar analysis can be done for the reactant. Then, it can be discussed if the force increases the single bond character of the C=C isomerizable bond (which could favor the isomerization) in both the reactant or the TS. Moreover, as can be seen in Figure S26, the TS is reached before 90 degrees once the force is applied. Can you argue this finding regarding the lowering of the energy barrier?

7) Line 197, in figure caption: in my pdf version the geometries corresponding to a 0.5nN force is not yellow but kind of pink-light orange.

Reviewer #1 (Remarks to the Author):

Comment. The manuscript presented by Craig and coworkers reports an up to 10000-fold thermal E-to-Z isomerization rate when a metal binding on the remote receptor of a molecular probe (stiff stilbene). Moreover, they perform DFT calculations to simulate the strain transportation process and discuss the relation between distance, dihedral angle, force, compliance, and energy difference (strain energy, isomerization energy, and transition energy barrier). Based on our research on related systems from the research team's prior publication record, there are three closely related papers using such stiff stilbene diphosphine system: 1) in Pd-catalyzed C-C bond coupling reactions (Angew 2014); 2) in Pt-induced reductive elimination (Chem Sci 2021) and 3) in Pd-induced oxidative addition (JACS 2020).

In this manuscript, the authors evaluate the mechanochemical efficiency of increasing strain energy to lower isomerization energy and the influence of binding affinity. The assumption is supported by kinetic studies and DFT calculations. We have to admit that the simulation design is well-thought and well-executed in this work. However, the same system has been studied in an author's previous paper Chem. Sci., 2021, 12, 11130 (doi: 10.1039/d1sc03182a). In that paper, they discuss the relationship between the mechanical force and the Pt complex's reductive elimination rate, where the force applied/stiff stilbene conformation relationship was well-established by both experimental and theoretical approaches. To some extent, this manuscript looks very similar to that (reversed story, how compressive force influences the E-Z isomerization in this paper vs. how E-Z isomerization affects the compressive force imposed on diphosphine ligand and thus on the rate of reductive elimination). And the conclusion or message of this paper is interesting but not general. It's hard to foresee how other scientists can build on this work. Thus in my view, this manuscript is not sufficiently novel and general for publication in Nature Communications.

Response. There are two aspects in which the results reported in the current paper speak to the (necessarily subjective) questions of "novelty and generality" in ways that we believe extend meaningfully beyond anything reported previously by us or anyone else:

1. Demonstration of allosteric lowering of an activation barrier below that of free substrate of a practically significant extent (100 – 10000-fold vs. the maximum of 3-5 fold hitherto demonstrated).
2. Demonstration that mechanochemical formalism provides an internally coherent and broadly generalizable approach to quantitative analysis of allostery. We are aware of no other articulated approach for analysis of allosteric accelerations (potentially due in part to the dearth of tractable examples of such accelerations); nor any prior recognition that mechanochemistry and allostery are conceptually linked.

Any assessment of the generality of a new interpretational approach is necessarily subjective. In the context of our paper it might be useful to acknowledge that molecular understanding of allostery has been suggested to be critical for catalysis, emergent properties, molecular machines and signal amplification.

The similarity between the papers cited by the reviewer and the current work is limited to the use of similar SS-based macrocycles as ligands to a metal. The cited papers identified new mechanochemical reactions and rationalized the measured mechanochemical kinetics. They teach us nothing about allostery, either conceptually, empirically or methodologically. The current work does all three.

The reviewer's comment suggest that our initial draft didn't make these points explicitly enough, and we have revised the paper as follows.

In the Introduction: "We are unaware of previous use of a mechanochemical formalism to support quantitative analysis of allosteric accelerations, which here provides insight into the structural regime where allosteric effects are greatest."

In the Conclusion: "For example, the dependence of ΔG^\ddagger on force shown in Fig. 2c quantifies the importance of molecular compliance as a design feature beyond simple change in geometry. The same plot also highlights the existence of "sweet spots" in applied force where the allosteric sensitivity is greatest; access to higher force regimes does not necessarily lead to greater allosteric regulation. Consequently, the mechanochemical formalism may prove valuable for guiding the design of synthetic allosteric catalysts and for quantitative tests of molecular models of allosterically controlled enzymatic activity as resulting from structural transmission of molecular strain across suitably stiff portions of the biomolecular scaffolds.⁵"

Comment. Here are some comments to help the author improve their manuscript:

1. A general scheme illustrating how the chemistry reported here to other general phenomena or science problems would help the reader to grasp the essence authors wanted to express.

Response. In response to this and other comments, we have revised Figure 1.

2. In Figure 1, The author shows Z(m,n) can isomerization to E(m,n) under 365 nm light radiation. However, once forming [Z(m,n)]PtCl₂, will the strain coming from the metal center prevents the stiff stilbene from isomerizing under the current radiation condition?

Response. We have not yet been able to switch from E to Z once the metal is coordinated.

3. The calculation in this paper is fruitful. However, there are too many physical quantities in this paper (>10), which makes the paper confusing and difficult to follow. I recommend listing all the quantities in the ESI and specifying their meaning, how they are defined, and calculation methods. This would definitely help readers to closely follow the discussion of the change of all the parameters.

Response. We appreciate the suggestion and believe that are terms are now clearly defined. The symbols used as parameters of the mechanochemical model are listed in Table S3. Standard ones, e.g., ΔG , are explained in the main text at first appearance. If there are specific terms still unaddressed, please let us know.

4. It seems like the calculation energy fits well with the experiment results, while the calculation level (u)BMK/def2SVP/(u)B3LYP/def2SVP (line 90) is not good enough for calculations in this system. There are mainly three questions: a) Why change functional B3LYP to an unpopular functional BMK? Has the author done any benchmarks about different functionals in your system? b) It is strange that the optimization of such organometallic compounds using B3LYP without empirical dispersion, have the authors attempted to add the empirical dispersion to improve the structural optimization accuracy (and it's nearly free for calculation)? c) For a system in this scale (~100+ atoms), the single-point calculation level is commonly upgraded to 3- ζ (e.g. def2-TZVP for Ahlrichs basis sets). The author might need to recalculate the single-point energy to see whether more accurate energy they can achieve.

Response.

- The selected model chemistry reproduced the measured ΔG^\ddagger of Z-E isomerization of SS(OMe)₂ and E-Z isomerization of diverse strained stiff-stilbene macrocycles with the lowest error (Table S4) among 7 functionals assessed.
- We reoptimized the minimum-energy conformers of (S,R)-E(2,2)PtCl₂ and (S,R)-E(2,2) and the corresponding isomerization transition states with B3LYP-d3/6-31+G(d) and summarized the results in the table below. The marginal difference between the relative energies of these conformers optimized with (B3LYP-d3) and without (B3LYP) dispersion suggest low likelihood that the inclusion of the dispersion would materially change the analysis of our experiments or the conclusions.

Table: The energies of the minimum-energy conformer of the isomerization transition state of (S,R)-(2,2) and (S,R)-(2,2)PtCl₂ macrocycles relative to the energy of the minimum-energy conformer of the E isomer, (S,R)-E(2,2) and (S,R)-E(2,2)PtCl₂ at (u)BMK/def2SVP/(u)B3LYP/def2SVP and (u)BMK/def2SVP/(u)B3LYP-d3/def2SVP levels

	B3LYP	B3LYP-d3
(S,R)-(2,2)	27.3	27.5
(S,R)-(2,2)PtCl ₂	21.2	22.2

- The def2TZVP-basis-set energy of the lowest-energy conformers of the isomerization transition state of the (S,R)-PtCl₂ macrocycle is 22.1 kcal/mol higher than that of the E isomer, vs. the 21.0 kcal/mol difference with the def2SVP basis set; for the corresponding Pt-free E(2,2) pair, this difference is 0.5 kcal/mol. Neither difference is large enough to change either the analysis of our experimental measurements or their conclusions. This suggests that spending ~85,000 cpu-h to re-calculate SPEs of all thermally-accessible conformers of all macrocycles with def2TZVP (6 macrocycles with 7 kinetically significant stationary states each, made of 5 thermally accessible conformers on average; def2TZVP corresponds to ~2500 – 3200 basis functions for our macrocycles with each SPE calculation at UHF requiring 200 – 500 cpu-h) would hardly be justified.

In summary, (u)BMK/def2SVP/(u)B3LYP/def2SVP reproduces measured isomerization ΔG^\ddagger accurately without needing unduly massive computational resources. We believe this addresses the substance of the reviewer's comments. However, if the reviewer is not satisfied with our answers, we respectfully ask him/her to clarify the following statements:

- The claims that the selected model chemistry both "fits well with the experiment results" and "is not good enough for calculations in this system" are mutually exclusive: if the selected model chemistry already reproduces all experimental quantities within experimental error what criterion does it fail to qualify as a poor choice for our macrocycles?
- We don't know why the reviewer characterizes BMK as "unpopular" or why it matters. The now-classical treatise by Truhlar (Quest for a universal density functional: the accuracy of density functionals across a broad spectrum of databases in chemistry and physics. *Philos. Trans. R. Soc.*, A 372, 20120476, 10.1098/rsta.2012.0476 (2014)) recommended BMK as one of 3 best functionals for calculations of reaction barriers, and since then BMK was recommended for several other types of energy calculations (e.g., 10.1021/acs.chemrev.5b00163; 10.1002/qua.26238; 10.1002/qua.25409). In other words, our choice of BMK is consistent with its good performance recorded in the literature for calculations of kinetic barriers.

3. We don't understand what the reviewer means by "more accurate energy": our calculations already reproduce measured standard ΔG^\ddagger to within the experimental uncertainty (the first 4 lines in Table 1, main text) making it impossible to reproduce the energies more accurately. Does the reviewer refer to some other reference energy that SPEs at def2TZVP and/or functionals with dispersion corrections would reproduce closer than those at def2SVP? If yes, we respectfully ask the reviewer to specify it and explain why that reference energy should be the preferred benchmark of accuracy over measured ΔG^\ddagger .

5. The calculation data is not fully provided. The coordinates and parameters (e.g. energy, force...) of binding affinity, mechanochemistry investigations, and transition state calculation should be supplemented.

Response. We listed the coordinates of the macrocycles as reactants (Z and E) and isomerization transition states (Appendix 1); the calculated free energies of activation are listed in Table 1 in the main text. We don't know what the reviewer means by coordinates of binding affinity, energy of binding affinity or force of binding affinity. The only quantum-chemically derived binding affinity used in our simulations are $\Delta\Delta G_{\text{bind}}$ of rxn 1 in the main text. These are listed in Table S5 and are calculated for each macrocycle, and as such are not a function of force. The procedure used for optimization of all structures, including SS(OMe)₂ coupled to a constraining potential, and the calculations of force-dependent energies (of SS(OMe)₂ and spring) for SS(OMe)₂ coupled to springs of different compliances are in the SI. We also added Tables S7-S9 with the quantitative implementation of the mechanochemical model of allosteric acceleration of E-SS isomerization (illustrative portions of this data are plotted in Figs. 2c-d and 3).

6. The term "reactor, receptor, spring, and effector" is confusing in this system. Thus, A figure showing the position of these components in the molecule is recommended to draw.

Response. We have revised Figure 1 and appreciate the suggestion.

7. line 152 "... compliance-independent at $\sim 0.2 \text{ \AA/nm}$." Does the author mean " $\sim 0.2 \text{ \AA/nN}$ "?

Response. Fixed.

8. line 191 "...yielding a range of $\langle f_E \rangle$ (0.35-0.6 nN) ." Is the value the same as line 189, "...both MeOC-COMe and O-O distances at f_E of 0.35 – 0.6 nN (Fig. 2b)" and f_E should be f_{TS} .

Response. The original notations are correct: line 189 refer to the geometry of a single conformer of E-SS(OMe)₂ (the one shown in Fig. 4a), whereas $\langle f_E \rangle$ refers to ensemble-average force; conceptually, for molecules coupled to a potential of finite compliance $\langle f_E \rangle$ and f_E are distinct, but in the specific case discussed in lines 188 – 195, the ensemble of E in this force range is dominated by the conformer shown in Fig. 4, making the single-conformer and ensemble ranges almost identical. The f_E should be f_E because only f_E , but not f_{TS} , is an experimentally exploitable control parameter: it's fairly straightforward to control the force exerted on a reactant (f_E) either by molecular design or micromanipulation techniques, whereas the force acting on the corresponding TS is not directly controllable.

9. Figure S26 in ESI, could the author add legends on these figures?

Response. Added.

10. Labels a and b are missing in Figure 4.

Response. Added.

11. The manuscript is not an easy read in general. One possible reason is that the authors used mainly very long sentences throughout. A bit of proofreading and a touch on the sentence structure would help.

Response. We have looked through the text with the benefit of time since submission and edited the paper with the reviewer's comments in mind. If there are specific passages that remain challenging, we would be happy to have those pointed out.

Reviewer #2 (Remarks to the Author):

In their manuscript "Allosteric control of olefin isomerization kinetics via remote metal binding and its mechanochemical analysis" Yu et al. describe how Pt coordination to a (S)-BIPHEP unit in a photoswitchable macrocycle affects the thermal E->Z double bond isomerization of a stiff stilbene unit.

In the first section of the manuscript, the authors provide experimental data for the isomerization kinetics of macrocycles with three different ring-sizes (E(2,2), E(2,3), E (3,3)) in their complexed and un-complexed form. Generally speaking, the activation free energies increase with increasing ring-size and decrease upon metal coordination. In the second section, an in-depth mechanochemical model for the observed allosteric acceleration of the thermal olefin isomerization is discussed. The theoretical model implies that a compressive force acting on the stiff stilbene unit is causing the rate acceleration of the reaction. The experimental and computational methods applied in this study seem sound.

In my opinion, a detailed mechanochemical description for the influence of a compressive force on the reactivity of a molecular entity is interesting and the fact that a small increase in strain accelerates thermal E->Z olefin isomerization dramatically is remarkable. The insights this study provides are important for various areas in chemistry and materials science.

While I think that the authors found an elegant system to describe the effect of compressive force on the double bond isomerization of stiff stilbene, I still have a few comments that should be addressed before publication:

- The readership of Nature Communications is very broad and some parts of the manuscript are a bit difficult to follow. It would be nice to show the general concept of the paper in figure 1 in a simplified manner. A schematic representation how the coordination of Pt applies a compressive load on the stiff stilbene unit would, in my opinion, improve the comprehensibility of the paper a lot.

Response. We appreciate the suggestions and have revised Figure 1 accordingly.

- The authors mainly discuss how metal coordination accelerates olefin isomerization by

imposing a compressive load on stiff stilbene. A similar (and intuitive) mechanochemical effect seems to apply by changing the size of the macrocycle from (2,2) to (2,3) to (3,3). This should be discussed by the authors in more detail in the main text.

Response. We have edited the text as follows.

Page 3: “The rate of isomerization increases as the size of the macrocycle decreases, with the ordering $E(2,2) > E(2,3) > E(3,3)$. This order is aligned with the relative ring strain of the macrocycles, but it has been demonstrated previously that restoring forces of carefully chosen internal molecular coordinates, rather than relative energies, are often a better quantitative correlant of reactivity. As discussed below, the mechanochemical formalism captures quantitatively the variation in isomerization kinetics across the whole range of the macrocycles, whether metalated or metal-free (e.g., Fig. 2b).”

Page 6: “To estimate the forces responsible for the variation of the kinetic stability of E stiff stilbene across both the metalated and unmetalled macrocycles...”

- Related to the comment above, macrocyclic molecular switches (such as hydrazones, see e.g. 10.1021/jacs.8b07612 and 10.1021/jacs.2c05384) and machines (such as molecular motors, see e.g. 10.1002/anie.202104285, 10.1038/s41565-021-01021-z, 10.1021/jacs.2c02547 and reference 23) attracted quite some interested in the last couple of years. Understanding how macrocycle size affects the isomerization behavior of such systems from a mechanochemical point of view would be of great interest for these areas of research. By connecting their study to (at least) the studies above (the list of examples is by no means exhaustive), the authors would put their work in a broader scientific context and could diversify the relatively small number of references provided in the current version of the manuscript.

Response. We have edited the text as follows.

Page 10: “In addition, macrocyclic molecular switches^{33,34} and machines^{23,35-37} are being developed for, and utilized in, an increasingly diverse range of purposes.³⁸⁻⁴⁰ The formalism applied here might inform approaches to increasing the efficiencies of switching,²² energy storage⁴⁵ and work-generation in such systems.¹⁵”

- The thermal E->Z isomerization was followed by ³¹P NMR spectroscopy and the rate plots are provided in the SI but not the corresponding NMR spectra. In order to better evaluate the kinetic experiments, the authors should provide the respective ³¹P NMR spectra (either superimposed or stacked) as well.

We now include representative stack plots of ³¹P NMR spectra for the isomerization of [E(2,2)]PtCl₂ and (S,S)-E(3,3) as figures S26 and S27 in the revised SI. In addition, all ³¹P NMR spectra relevant to our kinetic studies will be deposited in the Duke Research Data Repository upon acceptance of this manuscript.

- To be more consistent and avoid confusion, the authors should indicate which diastereomer was formed for E(2,2) and E(2,3) and use the appropriate nomenclature throughout the manuscript.

Response. Fixed

Minor comments:

- p2, line 43 and 45: Photolysis should be changed to photoisomerization
- p3, line 75: Furthered should be changed to further
- p7, figure 4: The caption refers to a) and b), which are not shown in the figure

Response. Fixed

In conclusion, I think that the study is of high interest for the broad scientific readership of this journal and I therefore recommend publication in Nature Communications after the suggested changes.

Reviewer #3 (Remarks to the Author):

The work presented by Yu et al is a clear example of mechanochemistry as an efficient tool to modulate the thermal reactivity of some molecules, in this case the isomerization of stiff-stilbene. This study is the follow-up of a previous work of this research group, from a more computational and theoretical point of view. The used methods are explained in detail and can be applied to other systems. The article is well-written and all the conclusions are well supported by the results. The findings and rationalization show the key role of allostery, motivating its use as a tool to enhance the efficiency of some thermal reactions. For all these reasons, I recommend the publication of this work after some minor revisions.

1) In lines 43 and 45. The E isomers are not form by photolysis of the Z isomers, as no bonds are broken in this process. It is a photoisomerization completed after irradiation of the Z isomer.

Response. Fixed

2) In the footnote of Table 1: b) is missing.

Response. Fixed

3) Apart from rationalizing the effect of Pt coordination on the free-energies (table 1) it can also be discussed the differences between the (2,2), (2,3) and (3,3) macrocycles. For E(2,2) is the lowest one while the values of (2,3) and (3,3) are similar. Which is the reason behind this trend?

Response. Although smaller E macrocycles are generally more strained than larger analogs the correlation between macrocycle size and the activation barrier of a strain-relieving reaction is confounded by the conformational preferences of the linkers, which often limit thermally accessible conformations of (usually) the transition state. The effect is particularly significant for stiff stilbene isomerization, because the molecular geometries change so much. Calculations suggest that E(2,2), E(2,3) and E(3,3) are 19.9, 10.7 and 9.9 kcal/mol less stable than the corresponding Z isomers; the <1 kcal/mol difference in ground-state strain of the (2,3) and (3,3) macrocycles is likely too small to overcome the specific conformational preferences of the two sets of linkers.

4) Along the manuscript, please unify the nomenclature of E-stiff (or E stiff). Also, E and Z should be written in italic.

Response. We have unified the nomenclature to E-stiff stilbene. Regarding the use of italic type, we are following the ACS Style Guide convention in which italic type is used for E and Z “when they appear with the chemical name or formula.” In the case of E(2,2) or Z-stiff stilbene, we believe these cases are equivalent to compound numbers and broad classes of compounds, respectively, and do not meet the stated criteria. We will, however, happily defer to the journal’s own editorial guidelines on this point.

5) Why in Table S2 is missing the data for macrocycle (3,3)?

Response. The activation free energies of E→Z isomerization of macrocycles (S,S)-E(3,3), (S,R)-E(3,3), and their metalated analogs were determined at a single temperature and hence, activation enthalpies and entropies were not determined. Activation free energies for isomerization of (S,S)-E(3,3), (S,R)-E(3,3), [(S,S)-E(3,3)]PtCl₂, and [(S,R)-E(3,3)]PtCl₂ are reported in Table 1.

6) In line 161, when rationalizing the larger strain when Pt-coordinated. Could it be possible to compare C=C bond length for the reactant and TS structures of Pt-free and Pt-coordinated macrocycles (or different strengths of the applied force) to check if the isomerizable double bond is weakened (similar to Figure S26)? In the SI is given the analysis of the torsion for different force magnitudes for the TS structure. A similar analysis can be done for the reactant. Then, it can be discussed if the force increases the single bond character of the C=C isomerizable bond (which could favor the isomerization) in both the reactant or the TS. Moreover, as can be seen in Figure S26, the TS is reached before 90 degrees once the force is applied. Can you argue this finding regarding the lowering of the energy barrier?

Response. We have added a figure to the SI showing the dependence of C=C bond length on f_E and f_{TS} and the torsion on f_E . We are not aware of quantitative relationships between a specific structural distortion and change in the activation barrier height. One of the values of the mechanochemical formalism is that force is related to these structural distortions and can be used to extract quantitative kinetic behavior, such as Figure 2.

7) Line 197, in figure caption: in my pdf version the geometries corresponding to a 0.5nN force is not yellow but kind of pink-light orange.

Response. We have changed this to “light orange” as suggested.

Reviewers' Comments:

Reviewer #1:

Remarks to the Author:

The authors answered all my concerns raised. The revised manuscript is much improved, in terms of clarity and readability. I would approve the publication of this work in Nat. Commun. after addressing the following minor issues:

- 1) Please improve the resolution of Fig. 2 and 4, as they both look blurry.
- 2) There are editing marks on Figure 1 to be cleaned out.

Reviewer #2:

Remarks to the Author:

Yu et al. addressed all my comments and I am entirely satisfied with their answers. In my opinion, the manuscript is now much easier to follow and the study is put into a broader scientific context.

There is only one minor issue with the current version of the manuscript: The middle part of figure 1b is extremely pixelated (at least in the PDF version I received) and should therefore be changed.

Apart from that I am happy to recommend the revised version for publication in Nature Communications.

Reviewer #3:

Remarks to the Author:

The authors have addressed all the proposed comments and changed the maintext and SI accordingly to them.

For this reason, I have no further comments so, I accept the article in its current version.